# Hotspots of the stokes rotating circulation in a large marginal sea

Jianping Gan [1✉], Hiusuet Kung[1], Zhongya Cai[2], Zhiqiang Liu[3], Chiwing Hui[1] & Junlu Li[1]

Marginal seas, surrounded by continents with dense populations, are vulnerable and have a quick response to climate change effects. The seas typically have alternatively rotating layered circulations to regulate regional heat and biogeochemical transports. The circulations are composed of dynamically active hotspots and governed by the couplings between unique extrinsic inflow and intrinsic dynamic response. Ambiguities about the circulations' structure, composition, and physics still exist, and these ambiguities have led to poor numerical simulation of the marginal sea in global models. The South China Sea is an outstanding example of a marginal sea that has this typical rotating circulation. Our study demonstrates that the rotating circulation is structured by energetic hotspots with large vorticity arising from unique dynamics in the marginal sea and is identifiable by the constraints of Stokes Theorem. These hotspots contribute most of the vorticity and most of energy needed to form and maintain the rotating circulation pattern. Our findings provide new insights on the distinguishing features of the rotating circulation and the dominant physics with the objectives of advancing our knowledge and improving modeling of marginal seas.

[1] Center for Ocean Research in Hong Kong and Macau, Department of Ocean Science and Department of Mathematics, Hong Kong University of Science and Technology, Hong Kong, China. [2] State Key Laboratory of Internet of Things for Smart City and Department of Civil and Environmental Engineering, University of Macau, Macau, China. [3] Department of Ocean Science and Engineering, Southern University of Science and Technology, Shen Zhen, China. ✉email: magan@ust.hk

Global marginal seas are separated from the major ocean basins by topographic features and links to the adjacent oceans such as the Atlantic Ocean (e.g., Baltic Sea[1], Mediterranean Sea[2], Gulf of Mexico[3]), Indian Ocean (e.g., Red Sea[4]), or Pacific Ocean (South China Sea, SCS[5]). These marginal seas exhibit rotating circulation patterns that are forced by an inflow-outflow water exchange with the adjacent oceans through straits such as Skagerrak Strait (connecting to Baltic Sea), Gibraltar Strait (Mediterranean Sea), Florida Strait (Gulf of Mexico), Bab-al-Mandeb Strait (Red Sea), and Luzon Strait (SCS). Other forcings include the local wind forcing and the coupled internal dynamics processes mainly governed by local topography and thermohaline structures[6–10]. These marginal seas generally have a deep basin surrounded by a steep slope, and their circulation are regulated by coupled external-internal dynamics. To demonstrate and explain the dynamics, we use as our example of a typical marginal sea, the SCS, which is the largest marginal sea in the Southeast Asia (Fig. 1a) and is surrounded by countries with ~22% of the world's total population based on the data of World Population Prospects 2019 from United Nations (https://population.un.org/wpp/).

Forcing by the Southeast Asia monsoonal wind stress and water exchanges with the adjacent oceans through the straits along the periphery of the SCS (Fig. 1a) produce a vertically dependent rotating circulation. The rotating circulation is cyclonic, anticyclonic, and cyclonic (CAC) circulation in the upper (<750 m), middle (750–1500 m), and deep (>1500 m) layers, respectively (Fig. 1a)[10–16]. This three-dimensional circulation has a profound influence on pathways of water mass path[17] and the transports of energy and biogeochemical substances[18–20] in the SCS.

Identifying the rotating layered circulation was initially based on estimates and speculation for only parts of the basin, at certain depths, or across specific transects. The circulation was represented with respect to geopotential[12–14] or isopycnal[15,16] levels. Under the wind stress curl forcing and due to the path of the SCS throughflow[20,21], the CAC circulation in the SCS consists of complex small-scale circulations that are regulated by local dynamics. Furthermore, the CAC circulation's existence is based only on domain integration of vorticity, which is formulated as the Stokes' Circulation Theorem (see Eq. (1) in Methods section), over the entire basin and relative to geopotential levels (Fig. 1b–d). We found that the layered CAC circulation does not exist in an isopycnal coordinate system (Fig. 1d) where the Stokes' circulation has a different physical meaning with when the circulation is viewed at the geopotential levels[22,23]. Our results show that the difference of circulations defined in geopotential and isopycnal coordinates is particularly significant over the continental slope where change of isopycnal depth is dramatic. Similar problems in depicting the circulation also exist in other marginal seas[1–5].

It is the layered influx-outflux-influx of planetary vorticity through the Luzon Strait (LS, Fig. 1a) and the corresponding vortex squeezing or stretching in the SCS basin that cause the CAC circulation to form in the basin[10]. Although the coupled internal-external dynamics have recently been found to link to the vertical transport associated with the cross-slope motion and three-dimensional energy transport and transfer[23,24], the detailed dynamics that frame the CAC circulation in the SCS remains unknown.

We validated CAC circulation in the SCS with geostrophic currents derived from GDEM (Fig. 1c) and with both observations and geophysical fluid dynamics reasoning[10,25]. We also found that most of the global circulation models with the same or higher resolutions, such as CMEMS[26], OFES[27], HYCOM[28], and others (not shown) have not well captured the layered CAC

circulation in the SCS (Fig. 1c). These global models failed to simulate the exchanges between the SCS and the adjacent oceans. Yet these exchanges provided crucial external fluxes to couple

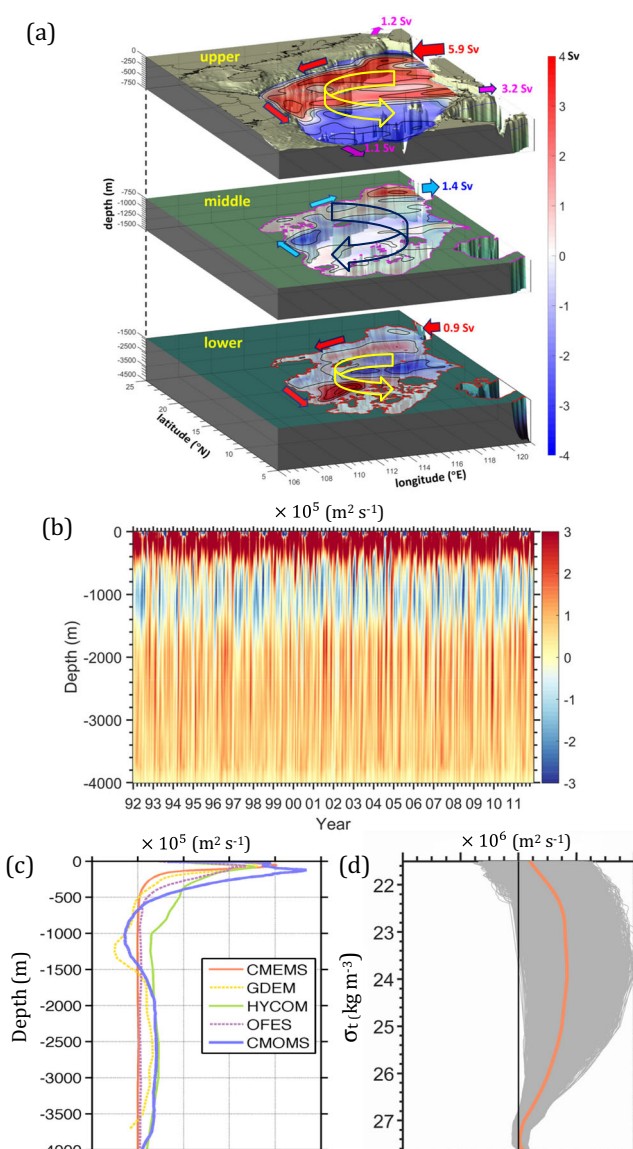

**Fig. 1 Characteristics of the layered circulation in the SCS. a** Schematic annual mean cyclonic-anticyclonic-cyclonic (CAC) circulation in the upper ($h \leq 750$ m), middle (750 m < $h \leq$ 1500 m) and deep layers ($h >$ 1500 m) of the SCS, respectively, based on CMOMS. The color contours represent transport (Sv) stream function. The arrows and transports (Sv) along the periphery of the SCS indicate the climatological mean volume transports through Taiwan Strait, Mindoro Strait, Karimata Strait, and Luzon Strait. **b** The 20-year time series of domain-integrated depth-dependent relative vorticity in the SCS basin. **c** Vertical profile of the 20-year time domain-integrated relative vorticity obtaining from geostrophic currents based on hydrographic data from GDEM (Generalized Digital Environmental Model; yellow dashed line), global models of CMEMS (The European Copernicus Marine Environment Monitoring Service; solid orange line), HYCOM (Hybrid Coordinate Ocean Model; green solid line) and OFES (Ocean General Circulation Model For the Earth Simulator; purple dashed line), and CMOMS (China Sea Multi-scale Modeling System; blue solid line) as a function of depth. Note, the data were averaged from 1991 to 2010. **d** Vertical profile of domain-integrated depth-dependent relative vorticity relative to isopycnal coordinates (gray lines) and their mean values (orange line) from 1992 to 2011.

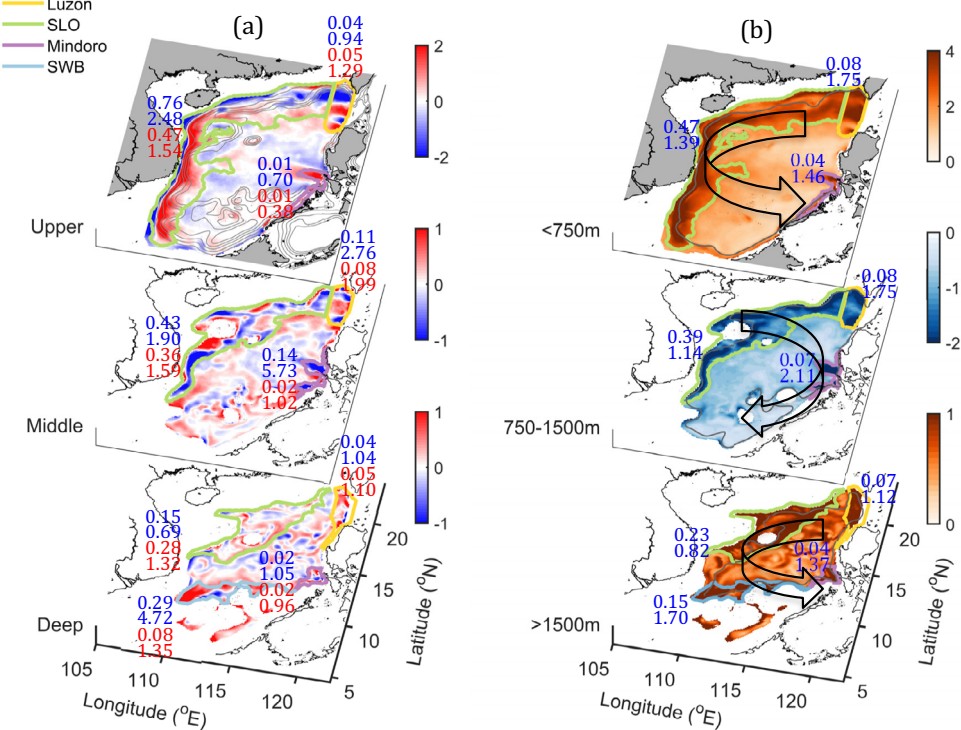

**Fig. 2 Net vorticity and hotspot vorticity in the layered circulation. a** Horizontal distribution of the depth-averaged vorticity ($10^{-5}$ s$^{-1}$) contributed to the formation of cyclonic-anticyclonic-cyclonic (CAC) circulation in the upper, middle, and deep layer. **b** Same as **a** but showing favorable vorticity ($\zeta_f$) to the CAC circulation. The hotspots include regions off Luzon Strait (LS), along the slope (SLO), near Mindoro Strait (MS) and in the southwest basin (SWB), which are bounded by the respective yellow, green, purple and blue thick contour lines. The regions of the hotspots are defined with vorticity thresholds of the 20-year mean values in the respective layer. The upper and lower values in each hotspot are $IR_{(M)}$ and $AR_{(M)}$ (blue), $IR_{(E)}$ and $AR_{(E)}$ (red) in **a** and $IR_{vort}$ and $AR_{vort}$ in **b**, which measure the contributions of mean kinetic energy (M), eddy kinetic energy (E) and vorticity (VORT) to hotspot, respectively. IR and AR are defined by Eqs. (2) and (3) in Methods section. The isobaths of the 200 m, 500 m, 1000 m, 1500 m, and 2000 m in the upper layer in **a** indicate the location of the slope. The 1500 m and 3000 m isobaths are shown in the middle and deep layers in **b**, respectively.

with internal dynamics for the formation of the CAC circulation. A similar modeling problem has also been reported for other marginal seas[29–31]. There are many reasons for this deficiency in the global model. A better understanding and thus better representation of circulation physics in the complex marginal sea such as flow-topography interaction internally and strait dynamics externally are certainly critical to improving of regional simulations in the global models, and it is the aim of this study to provide both.

## Results

In the SCS, the CAC circulation is strongly spatially variable and contributed mainly by vorticities ($\zeta$) in several key regions of the basin according to Stokes' Circulation Theorem (Fig. 2a). To distinguish these key regions of the CAC circulation from the other circulation, we defined a favorable vorticity ($\zeta_f$) that creates the CAC circulation for which $\zeta_f > 0$ in the upper and deep layers, and $\zeta_f < 0$ in the middle layer.

Figure 2b shows regions with relatively larger $\zeta_f$ that we recognized as hotspots for the CAC circulation. These hotspots are distinguished by $\zeta_f$ that is greater than $\zeta_f$ averaged over the entire domain for each layer. In this study, the hotspots are bounded by $\zeta_f > 2.2 \times 10^{-6}$ m s$^{-1}$, $\zeta_f < -1.1 \times 10^{-6}$ m s$^{-1}$, and $\zeta_f > 0.91 \times 10^{-6}$ m s$^{-1}$ in the upper, middle, and deep layers, respectively. In the end, we identified four distinct hotspots with strong $\zeta_f$. These hotspots are along the continental slope (SLO), off Luzon Strait (LS), off Mindoro Strait (MS), and in the southwestern basin (SWB) (Fig. 2b). Vorticity in these hotspots spatially matches the net vorticity averaged over each layer in

Fig. 2a, which indicates that the hotspots dominantly contribute most of the vorticity that forms the CAC circulation.

Figure 3a shows that the total $\zeta_f$ contributions of $IR_{vort}$ (see Eq. (2) in Methods section), from the hotspots to the basin CAC circulation are 58%, 55%, and 49% with the corresponding areas of 35.2%, 31.0%, and 33.3% in the upper, middle, and deep layers, respectively. Most of the contributions are from SLO (Fig. 2b) and the contribution along the western SLO to the CAC circulation is prominent (Fig. 2a). Thus, although the SLO occupies ~28.5%, 23.2%, and 19.1% of the basin area, the slope current flowing along the curved slope region has $IR_{vort}$ of 47%, 39%, and 23% in the upper, middle, and deep layers, respectively. Nevertheless, even though $IR_{vort}$ is relatively small for LS, MS, and SWB, because of the areas they occupy are ~4.6%, 2.6%, and 7.1% of the basin area, respectively, these hotspots are still dynamically active and important to the CAC circulation, as shown by their contributions per area or $AR_{VORT}$ values.

In addition to have a relatively large vorticity contribution to CAC as shown by $IR_{VORT}$ values, the hotspots are also characterized with active dynamics measured by $AR_{VORT}$ (see Eq. (3) in Methods section). $AR_{vort}$ reflects the intensity of the active dynamics or the contributions of $\zeta_f$ per unit area to the CAC circulation in each hotspot. In Figs. 2 and 3, all $AR_{vort} > 1$, which means the intensity of $\zeta_f$ for each hotspot is much stronger than for the rest of the basin. Unlike $IR_{vort}$, $AR_{vort}$ values in LS, MS, and SWB (Fig. 2b) are much larger than that in SLO, suggesting that the active dynamics in the CAC circulation are induced either by the exchanging flows between the SCS and the adjacent oceans through LS and MS or by flow over a topographic trough in SWB, respectively.

$\zeta_f$ in the hotspots is induced either by horizontal velocity shear in the jet-like slope current or by the curvature vorticity regulated

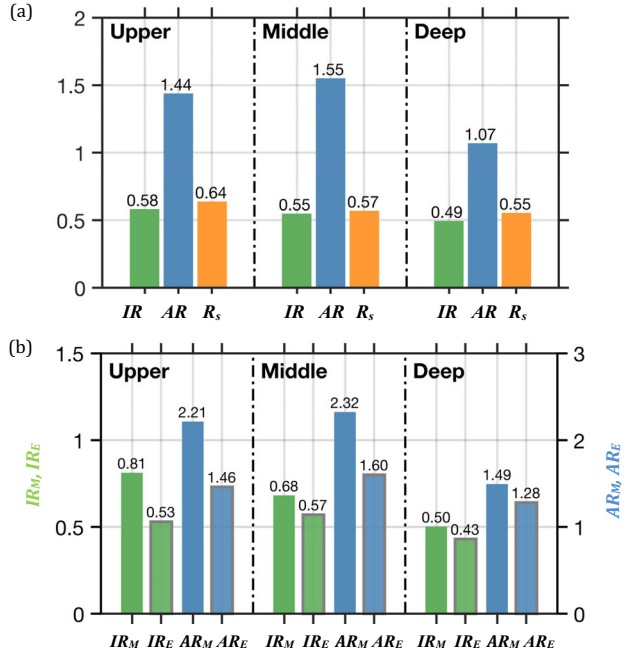

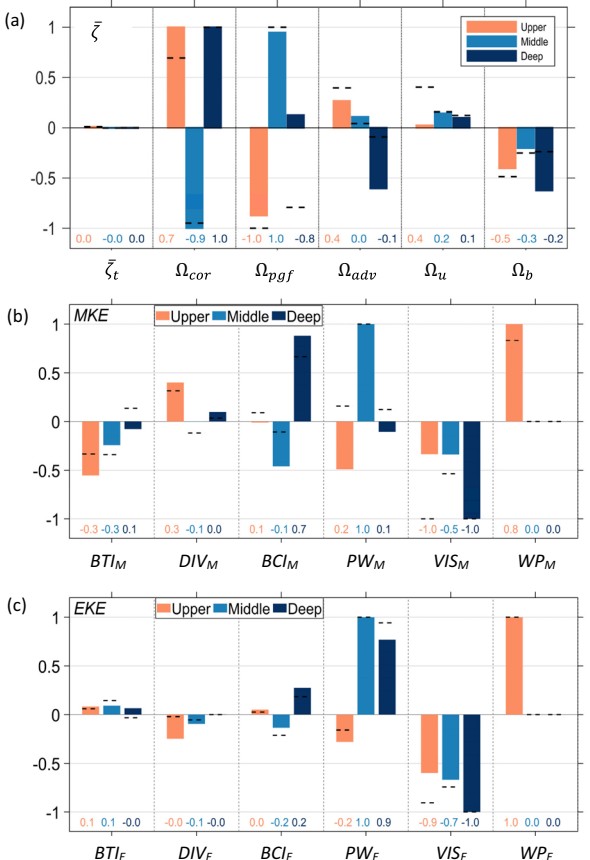

**Fig. 3 Relative and absolute contributions of vorticity and *MKE* by the hotspots. a** Annual mean values (bars and number) of $IR_{vort}$ (green), $AR_{vort}$ (blue), and ratio between shear vorticity and total vorticity $R_s$ (orange) in the hotspots for the three layers, respectively. **b** Similar to **a**, but for $IR_{(M,K)}$ and $AR_{(M,K)}$ of *MKE* and *EKE*, respectively.

by the curving isobaths along SLO. Figure 3a indicates that it is the shear vorticity that dominates the total vorticity in the hotspots. Our results (Supplementary Fig. 1) show that the relatively large contributions of shear vorticity are mainly found on the seaward side of the slope current along SLO, and the curving slope topography's contribution is less important contribution to $\zeta_f$. In addition, we found that the vorticity and dynamics that the hotspots contribute persist throughout the year even though the intensity of the CAC circulation is highly variable in time (Fig. 1b).

## Discussion

We investigated the vorticity balance in the hotspots to gain insight into the physics that determine how the CAC circulation forms in the SCS. Figure 4a shows the terms of the vorticity equations (see Eqs. (4) and (5) in Methods section) normalized within the hotspots. $\Omega_{cor}$ in all three layers predominantly contributes the vorticity to all the hotspots, which is indicated by $\Omega_{cor} > 0$ in the upper and deep layers and $\Omega_{cor} < 0$ in the middle layer. $\Omega_{cor}$ is related intrinsically to the stretching or squeezing of the water column along SL. In addition, $\Omega_{cor}$ extrinsically linked the influx of planetary vorticity (see Eq. (6) in Methods) into SLO in the upper and deep layers and the outflux in the middle layer[10]. Our analyses (Supplementary Fig. 2) showed that an additional contribution of positive $\Omega_{adv}$ in the upper layer forms in LS because of the westward influx of positive relative vorticity on the western side of the Kuroshio jet and because there is a larger kinetic energy (*KE*) gradient[22] (see Eq. (7) in Methods section). We see that $\Omega_{adv}$ is small in the other layers.

The balance between the bottom pressure torque $\left(\Omega_{pgf}\right)$ and $\Omega_{cor}$ in the hotspots reflects the intrinsic dynamics of the flow interacting with topography in response to the external influx or outflux across the boundaries of the hotspots. In the deep layer, the major sinks for the positive $\Omega_{cor}$ are nonlinear advection ($\Omega_{adv}$) and bottom stress curl ($\Omega_b$). A larger negative $\Omega_{adv}$ occurs in the deep layers of SWB, MS, and LS where an abruptly

**Fig. 4 Vorticity and *MKE* equations balances.** The 20-year mean of terms in Eq. (5) (upper panel), Eq. (10) (middle panel), and Eq. (11) (lower panel) in the upper (orange), middle (blue), and deep (black) layer of hotspots, respectively. In **a**, $\bar{\zeta}_t$ is acceleration, $\Omega_{cor}$ is divergence or vortex stretching, $\Omega_{pgf}$ is the bottom pressure torque, $\Omega_{adv}$ is horizontal advection and tilting, $\Omega_u$ and $\Omega_b$ are upper and bottom stress curl in each specific layer, respectively. **c** is similar to **b**, except for *EKE*. In **b**, **c**, $BTI_{(M,E)}$ are barotropic instability, $DIV_{(M,E)}$ are divergence of mean and eddy energy flux, $BCI_{(M,E)}$ are baroclinic instability due to mean or eddy buoyancy flux, $PW_{(M,E)}$ are horizontal mean and eddy pressure work, $VIS_{(M,E)}$ are turbulent viscosity dissipation and $WP_{(M,E)}$ are wind power input for *MKE* and *EKE*, respectively. Terms are normalized in hotspots. Black dashed lines and the corresponding numbers in each panel represent the corresponding normalized values over the entire SCS domain.

changing bottom topography forms stronger nonlinearities in the flow. $\Omega_b$ is negative in all layers and is relatively large in the upper and deep layers because of a stronger current and topographic influences, respectively.

The vorticity balance and underlying physics in the hotspots generally agree with the vorticity balance and physics for the entire basin[10], as shown by the dashed lines in Fig. 4a, which confirms the dominant role of the hotspots in forming the CAC circulation. However, some unique dynamics occurring within the hotspots deviate from the conditions occurring in the whole basin. For example, over an annual timescale, the wind stress curl imparts a net positive $\zeta_f$ for the cyclonic circulation in the upper layer over the entire basin[10]. However, in SLO where the $IR_{vort}$ is the largest, the negative wind stress curl in the northern basin during the winter balances the positive wind stress curl in the southern basin in the summer, and the annual net effect of wind stress curl on $\zeta_f$ in the hotspots is negligible. We identified that the net wind stress curl contributing to the annual mean CAC circulation comes mainly from the west of the Luzon islands

during winter. Another unique hotspot dynamics phenomenon is the presence of positive $\Omega_{pgf}$ in the deep layers of the hotspots. Our analyses (Supplementary Fig. 3) showed that the $\Omega_{pgf} > 0$ is induced by topographic troughs in SWB and LS where a relatively strong positive $\Omega_{pgf}$ occurs as flows climb up the troughs and squeeze the water column to balance the negative $\Omega_{cor}$. This finding suggests that through the follow-topography interaction, topography in a marginal sea can play an active, rather than just passive only, role in forming the CAC circulation.

The energy balance provides insight into how the hotspots contribute energy to the CAC circulation. The mean kinetic energy ($MKE$) from the hotspots reflects the strength of the mean circulation that maintains the CAC circulation while the terms of the $MKE$ equation (see Eqs. (9) and (10) in Methods section) provides the associated energy source and sink. Like the vorticity contribution from the hotspots, there is a large amount of $MKE$ in these hotspots in all layers[20], which further points to the active dynamic role of the hotspots have in the CAC circulation.

Figure 3b shows that in the upper layer, the hotspots substantially contribute $MKE$ ($IR_M = 81\%$) to the CAC circulation (Fig. 3b). Most of the $MKE$ (76%) occurs along SLO (Fig. 2a), indicating that the slope current primarily sustains the CAC circulation. The averaged $AR_M$ for the hotspots is 2.21 times larger than the average for the entire basin (Fig. 3b), ranging from ~0.7 in MS to 2.5 in SLO (Fig. 2a).

In the upper layer, the major contributors to the $MKE$ are $WP_M$ and $DIV_M$ (Fig. 4b). $DIV_M$ primarily occurs in LS in the upper layer, because of the kinetic energy influx from the Kuroshio intrusion. Major $WP_M$ exists in SLO in the upper layer and indicates the dominance of wind power input in driving the slope current. Unlike the small net wind stress curl contribution to the vorticity balance along SLO (Fig. 4a), the wind power input is the major kinetic energy source that drives and amplifies the slope current component of the CAC circulation. Wind power explains why the slope current in the SCS is 3 to 4 times greater than the volume of the Kuroshio intrusion through LS (~5.9 Sv, Fig. 1a)[25]. The strengthened slope current also enhances the basin-wide recirculation for maintaining the CAC circulation. $MKE$ in the upper layer is dissipated by $VIS_M$ and converted to eddy kinetic energy by $BTI_M$.

In the middle layer, there is a smaller, but still significant $IR_M$ (~68%) (Fig. 3b) in the hotsots, where SLO accounts for ~43% of the total $MKE$, and LS and MS, combined, have ~25% of the $MKE$ (Fig. 2a). Contributions of $AR_M$ are up to 2.32 times larger than the basin average in the middle layer, with very large contributions from MS and SLO: 5.73 times larger than the basin average in MS and 1.90 times larger than the basin average in SLO (Fig. 2a). Furthermore, along SLO and near MS and LS in the middle layer, mean pressure work $PW_M$ mainly induces $MKE$ and is balanced by barotropic $BTI_M$, baroclinic instability $BCI_M$, and $VIS_M$ (Fig. 4b). These results indicated that the extrinsic pressure flux and the intrinsic dynamic response are the major source and sink for the hotspot energy in the middle layer, respectively.

In the deep layer, although $IR_M$ is the smallest due to the relatively small geographic sizes of the hotspots, the hotspots are dynamically active enough to contribute 50% ($IR_M = 0.5$) to the CAC circulation in the basin. Interestingly, there is a larger contribution of energy in SWB where there is topographic trough (Fig. 3b). The mean $AR_M$ of the hotspots is 1.49, ranging from a very strong $AR_M$ of ~4.72 in SWB to a relatively small value of 0.69 in SLO (Fig. 2a).

The origin of $MKE$ in the hotspots of the deep layer is primarily from the strong circular current and a meandering current in SWB and SLO, respectively. The major source of the $MKE$ in this layer is generated by $BCI_M$ (Fig. 4b) or by potential energy released due to the vertical motion over the trough. The important role of topography is obvious in the deep layer, as is also shown by the larger $\Omega_{pgf}$ in SWB (Fig. 4a).

$EKE$ is also distinguished in hotspots, with significant contributions of $IR_E$ (0.53, 0.47, 0.43) and $AR_E$ (1.46, 1.60, 1.28) in different (upper, middle, deep) layers (Fig. 3b). The large values of $IR_E$ in all layers occur mainly along SLO, while the active local dynamics in other hotspots also generate significant $AR_E$ (Fig. 2a). Besides input by wind power $WP_E$ in the upper layer, $EKE$ is mainly contributed by internal pressure work and conversion from $MKE$ through internal $BCI_E$ due to active dynamics of the hotspots in both middle and deep layers (Fig. 4c). The distinguished $EKE$ in the hotspots shows their important contribution to the variability of the basin circulation.

This study shows the active dynamics of the hotspots that govern the characteristics, composition, and formation of the CAC circulation, and clarifies ambiguities regarding the physics of the CAC circulation in the SCS. The findings demonstrate an innovative concept for diagnosing the dominant physics of layered rotating circulation in marginal seas. By illustrating the importance of internal hotspots dynamics, and thus the coupled internal-external mechanism, we present the challenge in improving our physics comprehension and the associated modeling in the marginal seas.

## Methods

In our study, we used the China Sea Multi-scale Ocean Modeling System (CMOMS, https://odmp.ust.hk/cmoms/)[10,25]. The CMOMS model domain covers the northwest Pacific Ocean (NPO) and all the China Seas (SCS, East China Sea, Bohai Sea and Yellow Sea). The rectangular model domain extends from the southwest at (0.95°N, 99°E) to the northeast at (50°N, 145°E). The horizontal grid size of the model decreases gradually from ~10 km in the southern part to ~7 km in the northern part of the domain. Vertically, we adopted a 30-level stretched generalized terrain-following coordinate system.

We initialized the model with a mean hydrographic field from the Cross-Calibrated Multi-Platform (CCMP) dataset (ftp://podaac-ftp.jpl.nasa.gov/allData/ccmp/L3.0/flk). The model was spun up for a 50-year forcing by climatological atmospheric fluxes that had a 6-hourly frequency and 0.25° horizontal resolution. We integrated the CMOMS model forward from 1988 to 2011. Along the open boundaries of our model domain, we applied a novel open boundary condition that accommodated both tidal and subtidal forcing[32]. We obtained the tidal forcing along the open boundaries[33] and subtidal forcing from the Ocean General Circulation Model for the Earth Simulator (OFES) global model[27].

CMOMS is dynamically configured and numerically implemented in the SCS, and the model results are observationally and physically well-validated[10,25]. For our analyses, we used 20-year model variables from 1992 to 2011 that were averaged over 7 days at synoptic-scales.

Stokes' Circulation Theorem for the entire basin relative to geopotential levels is mathematically represented as

$$\Gamma = \oint \vec{\mathbf{V}}_{\mathbf{H}} \cdot \mathbf{dl} = \iint \nabla_H \times \vec{\mathbf{V}}_{\mathbf{H}} \, dA_{SCS}, \quad (1)$$

where $\Gamma$ is the basin circulation integrated along the boundary of the SCS basin with area $A_{SCS}$; $\vec{\mathbf{V}}_{\mathbf{H}}$ is horizontal velocity vector; and $\zeta = \nabla_H \times \vec{\mathbf{V}}_{\mathbf{H}}$ is the vertical relative vorticity normal to $A_{SCS}$.

$\zeta_f$ and the mean kinetic energy ($MKE$) contributions from our identified hotspots to the CAC circulation are measured by the relative contribution $IR_{(vort,M)}$, and by the intensity (contribution per unit area) of the active dynamics within the hotspots which are represented $AR_{(vort,M)}$,

$$(IR_{vort}, IR_M, IR_E) = \left( \frac{\iint \zeta_f dA_{hotspot}}{\iint \zeta_f dA_{scs}}, \frac{\iint MKE \, dA_{hotspot}}{\iint MKE \, dA_{scs}}, \frac{\iint EKE \, dA_{hotspot}}{\iint EKE \, dA_{scs}} \right). \quad (2)$$

$$(AR_{vort}, AR_M, AR_E) = \left( \frac{\frac{1}{A_{hotspot}}\iint \zeta_f dA_{hotspot}}{\frac{1}{A_{scs}}\iint \zeta_f dA_{scs}}, \frac{\frac{1}{A_{hotspot}}\iint MKE \, dA_{hotspot}}{\frac{1}{A_{scs}}\iint MKE \, dA_{scs}}, \frac{\frac{1}{A_{hotspot}}\iint EKE \, dA_{hotspot}}{\frac{1}{A_{scs}}\iint EKE \, dA_{scs}} \right). \quad (3)$$

These parameters determine the dynamic characteristics of the hotspots in the CAC circulation.

We expressed the vorticity equation, integrated by volume ($V$) over a specific layer in the hotspots, as[8],

$$\bar{\zeta}_t = \Omega_{cor} + \Omega_{adv} + \Omega_{pgf} + \Omega_u - \Omega_b + \Omega_h, \quad (4)$$

and Eq. (4) is fully expressed as

$$\int_V \overline{\xi_t}\,dV = -\overbrace{\int_A [(f\bar{u}D)_x + (f\bar{v}D)_y]\,dA}^{\Omega_{cor}} - \overbrace{\int_A \nabla_H \times \mathbf{NL}\,dA}^{\Omega_{adv}} + \overbrace{\int_A -\frac{1}{\rho_0}[(J(Lb,P^{Lb}) + J(Lu,P^{Lu}))]\,dA}^{\Omega_{pgf}}$$

$$+ \overbrace{\int_A \nabla_H \times \frac{\tau^{\mathbf{u}}}{\rho_0}\,dA}^{\Omega_u} - \overbrace{\int_A \nabla_H \times \frac{\tau^{\mathbf{b}}}{\rho_0}\,dA}^{\Omega_b} - \overbrace{\int_A \mathbf{v_t^u} \times \nabla_H \eta_u\,dA + \int_A \mathbf{v_t^b} \times \nabla_H \eta_b\,dA}^{\Omega_h} \quad (5)$$

where

$$\Omega_{cor} = -\int_{Si} fu\,dSi - \int_{Sj} fv\,dSj, \quad (6)$$

and

$$\Omega_{adv} = -\int_{Si}\left[\overbrace{u\xi}^{relative\ vorticity\ flux} + \overbrace{\left(\frac{1}{2}u^2 + \frac{1}{2}v^2\right)_y}^{KE\ gradient}\right]dSi + \int_{Sj}\left[\overbrace{v\xi}^{relative\ vorticity\ flux} + \overbrace{\left(\frac{1}{2}u^2 + \frac{1}{2}v^2\right)_x}^{KE\ gradient}\right]dSj, \quad (7)$$

where $\overline{\zeta_t}$ is volume-integrated changing rate of vorticity. The terms $Lu$ and $Lb$ are the depths, and $\tau^u$ and $\tau^b$ are the stress at the top and bottom for each specific layer, respectively. $J$ is a Jacobian operator and $NL$ is the nonlinear advection. $P^{Lu}$ and $P^{Lb}$ are pressure at the top and bottom of each layer, respectively, and $D$ is the total water depth. $S_i$ and $S_j$ are are the meridionally and zonally oriented sections., which are positive outward of the enclosed SCS area. The first term on the right-hand side of Eq. (4) is the layer-integrated divergence $\Omega_{cor}$ or (vortex stretching/squeezing), which is the equivalent to planetary vorticity flux across the open boundary of the hotspot region[10]. $\Omega_{adv}$ is the sum of horizontal relative vorticity advection and tilting, where horizontal advection represents the combined effects of relative vorticity flux and the kinetic energy gradient across the boundary. $\Omega_{pgf}$ is the bottom pressure torque reflecting the interaction between the baroclinically induced pressure and the variable slope topography, and $\Omega_u$ and $\Omega_b$ represent upper and bottom stress curl in each specific layer, respectively. These stress curls specifically refer to wind stress and bottom frictional stress curls in the upper and deep layers, respectively. $\Omega_h$ represents horizontal surface ($\mathbf{v_t^u}$) and bottom ($\mathbf{v_t^b}$) velocity accelerations interacting with the respective surface elevation ($\eta_u$) and bottom topography($\eta_b$). We found that $\Omega_h$ is generally small and can be neglected. Equation (4) explicitly illustrates circulation physics for a semi-closed marginal sea such as the SCS by distinguishing the coupled intrinsic and extrinsic dynamics for the CAC circulation.

We define $MKE$ and $EKE$ as[24]

$$MKE = \frac{1}{2}\rho_o\left(\bar{u}^2 + \bar{v}^2\right), \ EKE = \frac{1}{2}\rho_o\left(\bar{u'^2} + \bar{v'^2}\right), \quad (8)$$

where $(\bar{u}, \bar{v})$ are the time mean of the horizontal velocity and $(u', v')$ indicate their eddy states. The $MKE$ and $EKE$ equations are expressed as

$$(MKE, EKE)_{acce} = DIV_{(M,E)} + PW_{(M,E)} + BTI_{(M,E)} + BCI_{(M,E)} + WP_{(M,E)} + VIS_{(M,E)}. \quad (9)$$

The subscript, $M$ and $E$ refers to $MKE$ and $EKE$, respectively, and Eq. (9) is fully expressed as[20]

$$\frac{\partial MKE}{\partial t} = -\overbrace{\nabla \cdot \left(\overrightarrow{\mathbf{V}}\mathbf{MKE}\right)}^{DIV\_M} - \overbrace{\overrightarrow{V} \cdot \nabla \overline{P_a}}^{PW_M} - \overbrace{\rho_0\left[\bar{u}\nabla \cdot \left(\overrightarrow{V'}u'\right) + \bar{v}\nabla \cdot \left(\overrightarrow{V'}v'\right)\right]}^{BTI_M} - \overbrace{g\overline{\rho_a}\bar{w}}^{BCI_M} + \rho_0\overrightarrow{\mathbf{V}}_\mathbf{H} \cdot \overrightarrow{\mathbf{VIS}}_\mathbf{H}$$

$$(10)$$

$$\frac{\partial EKE}{\partial t} = -\overbrace{\nabla \cdot \left(\overrightarrow{\mathbf{V}}\mathbf{EKE}\right)}^{DIV\_E} - \overbrace{\left(\overrightarrow{V'} \cdot \nabla P'\right)}^{PW_E} - \overbrace{\rho_0\left[u'\overrightarrow{V'} \cdot \nabla \bar{u} + v'\overrightarrow{V'} \cdot \nabla \bar{v}\right]}^{BTI_E} - \overbrace{g\overline{\rho_a'w'}}^{BCI_E} + \rho_0\overrightarrow{V'}_\mathbf{H} \cdot \overrightarrow{\mathbf{VIS'}}_\mathbf{H} \quad (11)$$

The vertical integration of the last term in Eqs. (10) and (11) give $\int_{layer} \rho_0 \overrightarrow{V}_H \cdot \overrightarrow{VIS}_H\,dz = \overbrace{\overrightarrow{V}_{Hs} \cdot \overrightarrow{\tau_s}}^{WP_M} + VIS_M$ and $\int_{layer} \rho_0 \overrightarrow{V'}_H \cdot \overrightarrow{VIS'}_H\,dz = \overbrace{\overrightarrow{V'}_{Hs} \cdot \overrightarrow{\tau'}_s}^{WP_E} + VIS_E$. $\overrightarrow{V}$ and $\overrightarrow{V'}$ are the mean and perturbed parts of the three-dimensional velocity $(u, v, w)$, respectively. $\rho_a$ is the spatial anomaly of the density such that $\rho_a(x,y,z,t) = \rho(x,y,z,t) - \rho_r(z)$. $\overrightarrow{VIS}_H = (\vec{\mathbf{i}}\,VIS_u + \vec{\mathbf{j}}\,VIS_v)$ represents the vertical viscosity. The horizontal viscosity is relatively small and is included in the vertical viscosity for simplicity. $\vec{V}_{Hs}$ is the surface horizontal velocity, and $\vec{\tau}_s$ and $\vec{\tau}'_s$ are mean and perturbed surface wind stress, respectively. $DIV_{(M,E)}$, are the divergence of mean or eddy kinetic energy flux. $PW_{(M,E)}$, are the mean or eddy horizontal pressure work. $P_a$ is the pressure anomaly. $BTI_{(M,E)}$, are $MKE$ or $EKE$ input from barotropic instability due to the energy transfer between the mean and perturbed flow by nonlinear interaction. $BCI_{(M,E)}$ are $MKE$ or $EKE$ input from baroclinic instability due to the mean or eddy vertical buoyancy flux. $WP_{(M,E)}$ are wind power input to $MKE$ and $EKE$, and $VIS_{(M,E)}$ are the turbulent viscosity dissipation of $MKE$ and $EKE$, respectively.

## Data availability
The CMOMS data are available at https://odmp.ust.hk/cmoms/. The data from CMEMS are available at https://resources.marine.copernicus.eu. The data from HYCOM are available at https://tds.hycom.org/thredds/catalog.html. The data from OFES are available at http://apdrc.soest.hawaii.edu/ESC/escdata_main.php and the data from GDEM are available at https://catalog.data.gov/dataset/global-gridded-physical-profile-data-from-the-u-s-navys-generalized-digital-environmental-model

## Code availability
The model that we used in this study is the code from the community model, ROMS, (https://www.myroms.org/). The code associated with this paper is available by request from J.G.

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

## Acknowledgements

This study was supported by the Key Research Projects 41930539 (J.G.), 41906016 (Z.L.), and 42006009 (Z.C.) of the National Science Foundation, China (NSFC). J.G. was also supported by the Center for Ocean Research in Hong Kong and Macau (CORE), a joint research center between Qingdao National Laboratory of Marine Science and Technology and Hong Kong University of Science and Technology, the Theme-based Research Scheme (T21-602/16 R) and the Hong Kong Research Grants Council (GRF 16212720). Z.C. was also supported by the Science and Technology Development Fund, Macau SAR (SKL-IOTSC(UM)-2021-2023). We are also grateful for the support of The National Supercomputing Centers of Guangzhou and Tianjin.

## Author contributions

J.G. conceived the study, performed the analyses, built the mechanism, and wrote the paper. H.K., Z.Y., Z.L., C.H., and J.L. contributed to processing data, improving the paper, and assisted in interpretating the results.

## Competing interests

The authors declare no competing interests.
