## [Peer Review File · Nature Communications]

Title: Hotspots of the Stokes Rotating Circulation in a Large Marginal SeaREVIEWER COMMENTS

Reviewer #1 (Remarks to the Author):

Review on “Hotspots of the Stokes Rotating Circulation in a Large Marginal Sea” [Paper # NCOMMS-21-37992-T]

The authors have done a nice job demonstrating their hypothesis that energetic hotspots with large vorticity contribute most for maintaining the rotating circulation pattern in the South China Sea. This study will provide new insights on not only dynamics of ocean circulation, but also knowledge for improving numerical simulation of the marginal sea in global ocean models. The manuscript is well organized and well written. I suggest a minor revision. Line by line comments are below.

Comments:

1. Line 70-73. “We found that the layered CAC circulation does not exist in an isopycnal coordinate system (Fig. 1c) ...”. The result show that there is no CAC circulation in CMEMS, HYCOM, and OFES, all of which are high resolution. Some previous studies with coarse resolution have, however, show CAC circulation in the SCS (e.g., Wei et al., 2016; Zhu et al., 2017). Is there any explanation?

Reference:

Wei ZX*, Fang GH, Xu TF, Wang YG, Lian Z. (2016). Seasonal variability of the isopycnal surface circulation in the South China Sea derived from a variable-grid global ocean circulation model. *Acta Oceanologica Sinica*, 35, 11–20. <https://doi.org/10.1007/s13131-016-0791-3>

Zhu Yaohua, Sun Junchuan, Wang Yonggang, Wei Zexun*, Yang Dezhou, Qu Tangdong, 2017. Effect of potential vorticity flux on the circulation in the South China Sea. *Journal of Geophysical Research: Oceans*, 122(8). 6454-6469, doi:10.1002/2016JC012375.

2. The eddy are also active along the northern slope of SCS, the authors suggest that the MKE is converted to eddy kinetic energy by BTIM. But, does the EKE contribute to the hotspots? I think the authors should better make more discussion on the role of eddies.

3. Strong mixing and Luzon Strait overflow are also important for maintaining the CAC circulation pattern. Could the authors give a comparison of contribution among the mixing, Luzon Strait overflow, Luzon Strait transport, and the hotpots?

4. There are too many abbreviations, I suggest only keep few necessary and common abbreviation.

5. Page 16, the section “Methods”, “Data availability” and “Code availability” are wrongly inserted between the reference-29 and 30

Reviewer #2 (Remarks to the Author):

Based on a regional numerical model with terrain-following coordinate covering the northwestern Pacific and the China Seas, the authors analyzed the cyclonic-anticyclonic-cyclonic (CAC) circulation in the South China Sea (SCS), revealing four active zones regulating the CAC circulation pattern relative to

the geopotential levels, i.e., the continental slope, west of the Luzon Strait, west of Mindoro Strait, and the southwestern basin. Energy balance analysis further indicated the source of energy for the active zones. Generally, the manuscript is well structured and brings further knowledge of interests to the community. However, as below, this manuscript has a few arguable points expecting to be examined by the authors.

Major concerns:

1. The regional model used in this manuscript, so called the CMOMS, is validated by GDEM climatology dataset with domain integrated relative vorticity, which shows the capability to resolve the CAC circulation pattern. On the contrary, those global reanalysis datasets and GCMs perform not well. Usually, due to the finer and more adaptive vertical level configuration, regional models would perform better than global models on simulating regional ocean circulation. So, what are the specific advantages of the CMOMS over other models making it suitable in simulating the SCS circulation?
2. Those four active zones revealed by the model, one of the main results of this manuscript, are tightly related the circulation pattern in each layer. On the surface layer, a lot of observations could be used on the validation of the results. But in the intermediate and deep layers, where observations are not that adequate, I suggest the manuscript clarify in details how to confirm that the simulated circulation is reliable.
3. As introduced in the beginning, this work is focused on the dynamics which is critical to improving regional simulation in global models. It is fine to use the SCS as an example. But how could the results of this manuscript contribute to the regional simulation should be further clarified.

Detailed comments:

Line 19: Delete '(SCS)'

Line 27-29: Maybe the Indian Ocean and the corresponding marginal sea like the Red Sea should also be listed here.

Line 38: '~22% of the world's total population' is a specific number which should be based on some references.

Line 39: Delete "all"

Line 66-67: Energetic internal waves, mesoscale processes, etc., could also contribute significantly on forcing the regional circulation.

Line 71: Fig.1c should be Fig.1d?

Line 81: 'validated our simulated SCS CAC circulation' should be 'validated our model and other models, such as CMEMS, OFES, HYCOM, with ...'

Line 84-85: The performance of GCMs in intermediate and deep layers around the global ocean is yet a big issue. We cannot yet say these models performed well in open oceans.

Line 92: It should be clarified how the CAC circulation be 'well-regulated by vorticities in several key regions' based on Fig.2a.

Line 94: 'Favorable vorticity' should be defined.

Line 98: The four regions identified in Fig.2 generally accounts for half of the basin, which can hardly be identified as 'hotspots'. Besides, IRvort of the four regions, covering 50% of the whole area, only accounts for 58% of the whole domain.

Line 141: Dynamics regulating the circulation are complicated. It's not necessarily 'unique'. I suggest remove the sentence.

Line 143: Fig.3->Extended Data Fig.1

Line 151-170: Since the energetic domains are important, perhaps the comparison between those four domains and the other areas should be discussed here.

Line 180: annual mean CAC circulation.

Line 188: energy balance analysis

Line 246: 'observed'->'identified in other model'

Fig.1 It's difficult to distinguish the isobaths and streamfunction in a).

Fig.2 Labels of a and b are missed. Also, it's difficult to distinguish the lines from the background color.

And IR an AR should be defined.

Fig.3 Label of a should be added.

As a minor note, I found some grammatical and layout errors in the main text and reference list that I am not all listing here. I suggest proper editing of the manuscript.

Response to Reviewer 1

Reviewer #1 (Remarks to the Author):

Review on “Hotspots of the Stokes Rotating Circulation in a Large Marginal Sea” [Paper # NCOMMS-21-37992-T]

The authors have done a nice job demonstrating their hypothesis that energetic hotspots with large vorticity contribute most for maintaining the rotating circulation pattern in the South China Sea. This study will provide new insights on not only dynamics of ocean circulation, but also knowledge for improving numerical simulation of the marginal sea in global ocean models. The manuscript is well organized and well written. I suggest a minor revision. Line by line comments are below.

Response: We appreciate the time and help of the reviewer.

Comments:

1. Line 70-73. “We found that the layered CAC circulation does not exist in an isopycnal coordinate system (Fig. 1c) ...”. The result show that there is no CAC circulation in CMEMS, HYCOM, and OFES, all of which are high resolution. Some previous studies with coarse resolution have, however, show CAC circulation in the SCS (e.g., Wei et al., 2016; Zhu et al., 2017). Is there any explanation?

Reference:

Wei ZX*, Fang GH, Xu TF, Wang YG, Lian Z. (2016). Seasonal variability of the isopycnal surface circulation in the South China Sea derived from a variable-grid global ocean circulation model. *Acta Oceanologica Sinica*, 35, 11–20. <https://doi.org/10.1007/s13131-016-0791-3>

Zhu Yaohua, Sun Junchuan, Wang Yonggang, Wei Zexun*, Yang Dezhou, Qu Tangdong, 2017. Effect of potential vorticity flux on the circulation in the South China Sea. *Journal of Geophysical Research: Oceans*, 122(8). 6454-6469, doi:10.1002/2016JC012375.

Response: ‘...the Stokes’s circulation in isopycnal coordinate has a different physical meaning from when the circulation is viewed at the geopotential levels^{20,23}.’, as stated in the paper. Fig. R1 below shows the definition of circulation based on geopotential and isopycnal coordinates from Cai et al. (2021)²³.

Fig. R1. Conceptual diagram of the vorticity, ξ , and basin circulation, Γ , in geopotential (with subscript z) and isopycnal (with subscript ρ) levels. ξ_{ρ}^z represents the vertical component of the relative vorticity (ξ_{ρ}) in the isopycnal level (from Cai et al., 2021²³).

As result of the physical meaning of circulation in different coordinates, we found that ‘The difference of circulation defined in geopotential isopycnal coordinates is particularly significant over the slope where change of isopycnal depth is dramatic.’ This information is now improved and further clarified in the revised ms.

References

²⁰. Cai, Z., & Gan, J. (2019). Coupled external-internal dynamics of layered circulation in the South China Sea: A modeling study. *J. Geophys. Res. Oceans*, 124, 5039-5053.

²³. Cai, Z., Gan, J., Liu, Z., Hui, R., & Li, J. (2020). Progress on the formation dynamics of the layered circulation in the South China Sea. *Progress in Oceanography*, 181(202).
<https://www.sciencedirect.com/science/article/pii/S0079661119304264>

‘There are many reasons for this deficiency in the global model. A better understanding and thus better representations of the complex marginal sea physics such as flow-topography interaction internally and strait dynamics externally are certainly critical to improving of regional simulations in the global models, and it is the aim of this study to provide both.’

The formation of CAC circulation is mainly determined by the external forcing of planetary vorticity influx/outflux and wind forcing (for the upper ocean) and internal dynamic responses. The model resolution is not as important as coupled external forcing and internal dynamics response captured by the model. For external forcing, the correct simulation of the transport through straits, particularly regarding to the inflow-outflow-inflow vertical structure in Luzon Strait, is critical. For internal dynamics, the slope topography and the associated dynamics is the key, as shown by the effect of the slope hotspot in this study. The transport through Luzon Strait is critically determined by correct simulation of Kuroshio, circulation system in the western Pacific Ocean as well as the coupling internal circulation in the marginal sea.

We have further elaborated this in the revised paper.

Wei et al. (2016) is a modeling study while Zu et al. (2017) used P-vector inverse method to derive circulation in the SCS. Both did not use Stokes Theorem to represent the basin-wide circulation. In addition, they calculated vorticity in isopycnal coordinate at which vorticity or circulation has different physical meaning from geopotential coordinate. We argued that it is physically sensible to define vorticity in the direction normal to geopotential level when we define the CAC layered circulation (also see Response above).

2. The eddy are also active along the northern slope of SCS, the authors suggest that the MKE is converted to eddy kinetic energy by BTIM. But, does the EKE contribute to the hotspots? I think the authors should better make more discussion on the role of eddies.

Response: Yes, *EKE* contributes mainly to the variability of the mean circulation in the hotspots and it sustains the mean circulation through energy conversions between *EKE* and *MKE*. We add a paragraph discussing characteristics and role of *EKE* in “Energy sources in the hotspots”. We also include *EKE* in Figures 2a, 3 and 4.

3. *Strong mixing and Luzon Strait overflow are also important for maintaining the CAC circulation pattern. Could the authors give a comparison of contribution among the mixing, Luzon Strait overflow, Luzon Strait transport, and the hotpots?*

Response: It is known that stronger mixing in the SCS than in the western Pacific Ocean can create density difference, and thus produces a *net* pressure gradient force (Gan et al., 2006) for the intrusion of current from the western Pacific Ocean into the SCS through Luzon Strait, particularly in the deep basin (Qu et al. 2006). The intrusion flows along the slope and provides planetary vorticity for the formation of the CAC circulation as introduced in this paper and Gan et al. (2016).

In fact, the dynamics of the western boundary current (Kuroshio), mixing and hydraulic control of the Luzon Strait all control the intrusion. The mixing itself in the SCS is part of the consequences of the interaction between the CAC circulation and the marginal sea dynamics, and it is hard to be called as a “driver” although the mixing certainly provides feedback for the circulation. These complex dynamics need to be further investigated in separated paper. The current study investigated the hotspot dynamics of the CAC circulation or the unknown characteristics and dynamics of the CAC circulation inside of the SCS as a result of various forcing such as wind forcing, slope current due to intrusion and wind, flow-topography interaction. The study has included the effects of the intrusion and the overflow induced by various physical forcing.

References

⁵ Gan, J., H. Li, E. N. Curchitser and D. B. Haidvogel, 2006. Modeling South China Sea circulation. Response to seasonal forcing regimes. *J. Geophys. Res. (Oceans)*, 111, C06034, doi:10.1029/2005JC003298.

Qu, T., J. B. Girton, and J. A. Whitehead (2006), Deepwater overflow through Luzon Strait, *J. Geophys. Res.*, 111, C01002, doi:10.1029/2005JC003139.

4. *There are too many abbreviations, I suggest only keep few necessary and common abbreviation.*

Response: Most of the abbreviations are locations and commonly used physic variables. We have reduced the number in the revised manuscript.

5. *Page 16, the section “Methods”, “Data availability” and “Code availability” are wrongly inserted between the reference-29 and 30.*

Response: We followed the format of the journal by separating the references cited in the main text and outside the main text.

Reviewer #2 (Remarks to the Author):

Based on a regional numerical model with terrain-following coordinate covering the northwestern Pacific and the China Seas, the authors analyzed the cyclonic-anticyclonic-cyclonic (CAC) circulation in the South China Sea (SCS), revealing four active zones regulating the CAC circulation pattern relative to the geopotential levels, i.e., the continental slope, west of the Luzon Strait, west of Mindoro Strait, and the southwestern basin. Energy balance analysis further indicated the source of energy for the active zones. Generally, the manuscript is well structured and brings further knowledge of interests to the community.

Response: We appreciate the time and help of the reviewer.

However, as below, this manuscript has a few arguable points expecting to be examined by the authors.

Major concerns:

1. The regional model used in this manuscript, so called the CMOMS, is validated by GDEM climatology dataset with domain integrated relative vorticity, which shows the capability to resolve the CAC circulation pattern. On the contrary, those global reanalysis datasets and GCMs perform not well. Usually, due to the finer and more adaptive vertical level configuration, regional models would perform better than global models on simulating regional ocean circulation. So, what are the specific advantages of the CMOMS over other models making it suitable in simulating the SCS circulation?

Response: There are many reasons for this deficiency in the global models, as pointed out in the paper. A better understanding and better representation of the complex circulation physics are certainly critical to improving of regional simulations in the global models, and it is the aim of this study to provide both.

Although it would require a separated study to thoroughly provide answer for the failure of global models in capturing the CAC circulation in the SCS, we summarize the following major reasons as presented in the paper:

1: The resolution is not a major issue since the global circulation models, such as CMEMS24, OFES25, HYCOM26, and others (not shown), have the same or higher resolutions as compared with our model, as mentioned in the paper.

2: Better representation of the complex marginal sea physics is critical, such as flow-topography interaction internally and strait dynamics externally as demonstrated by hotspots dynamics in this study. The ocean circulation in marginal sea is forced not only by wind and intrusion from the adjacent seas externally, but also by intensive dynamics processes involving flow-topography interaction internally. More importantly, the internal and external forcings couple each other to control both the circulation in marginal sea and water exchange between the sea and adjacent ocean. The global models did not well simulate the exchanges between the SCS and the adjacent oceans.

The global models did not well capture such as slope topography effect of the marginal sea and straits linking the sea and adjacent ocean. These may be attributed to, for example, misrepresentation of topography by vertical coordinate (e.g. by z -coordinate), twisted circulation dynamics by over-usage of data assimilation, and incoherent dynamics linkage among the adjacent ocean and sea.

A physics and physical-accommodated investigation of global models require a separate investigation, and we demonstrated this important issue by conducting “hotspots” analyses in this paper.

2. Those four active zones revealed by the model, one of the main results of this manuscript, are tightly related the circulation pattern in each layer. On the surface layer, a lot of observations could be used on the validation of the results. But in the intermediate and deep layers, where observations are not that adequate, I suggest the manuscript clarify in details how to confirm that the simulated circulation is reliable.

Response: In addition to the newly added evidence of layered circulation derived from GEDEM data, extensive validations of the SCS circulation by both observations and dynamics reasoning have been provided in Gan et al. (2016a) and Gan et al. (2016b). We improved these descriptions in Methods and added them briefly in the main text in the revised ms.

3. As introduced in the beginning, this work is focused on the dynamics which is critical to improving regional simulation in global models. It is fine to use the SCS as an example. But how could the results of this manuscript contribute to the regional simulation should be further clarified.

Response: The understanding of the circulation physics, such as hotspots dynamics in marginal sea in this study, is critical to improve the simulation of the marginal sea. We further strengthened the contribution in the revised ms,

‘This study shows the active dynamics of the hotspots that govern the characteristics, composition, and formation of the CAC circulation, and clarifies ambiguities regarding the physics of the CAC circulation in the SCS. The findings demonstrate an innovative concept for diagnosing the dominant physics of layered rotating circulation in marginal seas. By illustrating the importance of internal hotspots dynamics, and thus the coupled internal-external mechanism, we present the challenge in improving our physics comprehension and the associated modeling in the marginal seas.’

Detailed comments:

Line 19: Delete ‘(SCS)’

Done.

Line 27-29: Maybe the Indian Ocean and the corresponding marginal sea like the Red Sea should also be listed here.

We now include Red Sea which also exhibits layered exchange flow though the Bab-al-Mandeb strait and rotating circulation inside Red Sea. Relevant references are also added.

Line 38: ‘~22% of the world’s total population’ is a specific number which should be based on some references.

The reference of the data used to calculate the percentage is added.

Line 39: Delete “all”

Done.

Line 66-67: Energetic internal waves, mesoscale processes, etc., could also contribute significantly on forcing the regional circulation.

Agree, but the effect of these physical processes on the CAC circulation in the SCS needs further investigation and are not the focus of this study.

Line 71: Fig.1c should be Fig.1d?

Revised and thanks.

Line 81: ‘validated our simulated SCS CAC circulation’ should be ‘validated our model and other models, such as CMEMS, OFES, HYCOM, with ...’

We have revised text.

Line 84-85: The performance of GCMs in intermediate and deep layers around the global ocean is yet a big issue. We cannot yet say these models performed well in open oceans.

Agree and the sentence is removed.

Line 92: It should be clarified how the CAC circulation be ‘well-regulated by vorticities in several key regions’ based on Fig.2a.

The sentence is revised as “...contributed mainly by vorticities (ζ) in several key regions of the basin according to Stokes Circulation Theorem (Fig. 2a).”

Line 94: ‘Favorable vorticity’ should be defined.

Favorable vorticity (ζ_f) is defined as: “We defined a favorable vorticity (ζ_f) that creates the CAC circulation for which $\zeta_f > 0$ in the upper and deep layers, and $\zeta_f < 0$ in the middle layer.”

Line 98: The four regions identified in Fig.2 generally accounts for half of the basin, which can hardly be identified as ‘hotspots’. Besides, IRvort of the four regions, covering 50% of the whole area, only accounts for 58% of the whole domain.

The percentage of hotspot area we used was relative to the surface area and was not accurate. We now obtain the percentage by considering the changing area with depth in each layer,

$$R_{\text{hotspot}} = \frac{\sum \text{AREA}_{\text{hotspot}}(z)}{\sum \text{AREA}_{\text{SCS}}(z)}$$

Thus, percentages of hotspot area and their favorable vorticity contributions to the basin CAC circulation are revised as 58%, 55% and 49% with the corresponding areas of 35.2%, 31.0 % and 33.3 % in the upper, middle, and deep layers, respectively.

As described in Methods and also elaborated in main text in the revised manuscript, the hotspots are characterized by both relative contribution $IR_{(vort)}$, and the intensity (contribution per unit area) of the active dynamics $AR_{(vort)}$ within the hotspots. Thus, hotspots not only have apparently higher values of favorable vorticity (ζ_f), but also exhibit significantly active dynamics in structuring the CAC circulation, as highlighted in Fig. 2a,b.

For example, among 58% of $IR_{(vort)}$ in the upper layer, 47% was contributed over the slope with <28.5% of the total basin area. Meanwhile, $AR_{(vort)}$ are also significant. These two indexes, together with corresponding indexes for MKE and EKE , reflect the nature of hotspots.

We have revised the text with inclusion of more detailed information.

Line 141: Dynamics regulating the circulation are complicated. It's not necessarily 'unique'. I suggest remove the sentence.

Done.

Line 143: Fig.3->Extended Data Fig.1

Revised Fig. 3 to Fig.3a.

Line 151-170: Since the energetic domains are important, perhaps the comparison between those four domains and the other areas should be discussed here.

These contents have been discussed in the subsequent section Energy analysis in the hotspots

Line 180: annual mean CAC circulation.

Done.

Line 188: energy balance analysis

We keep the original subtitle since the section contains more than the balance.

Line 246: 'observed' -> 'identified in other model'

The sentence and paragraph have been revised.

Fig.1 It's difficult to distinguish the isobaths and streamfunction in a).

Only transport stream function is shown in the ocean of Fig. 1a by color contours. We now remove the colorbar for the water depth to avoid confusion.

Fig.2 Labels of a and b are missed. Also, it's difficult to distinguish the lines from the background color. And IR an AR should be defined.

Labels are added. IR and AR are defined, and we add relevant information in caption.

Fig.3 Label of a should be added.

Labes are added.

As a minor note, I found some grammatical and layout errors in the main text and reference list that I am not all listing here. I suggest proper editing of the manuscript.

We have gone through the entire paper, polished the language and check format and references.

REVIEWERS' COMMENTS

Reviewer #1 (Remarks to the Author):

The author has answered my questions and made necessary modifications to the manuscript. My suggestion is acceptable.

Reviewer #2 (Remarks to the Author):

In their revised version of the manuscript of 'Hotspots of the Stokes Rotating Circulation in a Large Marginal Sea', I am pleased that the authors responded properly to most of our reviews. This manuscript may be accepted after some minor corrections as below.

Line 40-41. Except for mentioning some other marginal seas when addressing the background, no more case study has been done outside of the SCS by this study. So, I suggest either add case study of other marginal seas, or remove this sentence.

Fig.1a&b. The model based on which these results are calculated should be indicated. E.g., '(a) Schematic annual mean cyclonic-anticyclonic-cyclonic ... of the SCS, respectively, based on CMOMS'.

Line 73 'than' -> 'with'

Line 78 'through the Luzon Strait'

Line 88-89. In my opinion, these models do not manage to simulate the intermediate and deep circulation well in the adjacent oceans. A lot discrepancies exist among these models and most of them fail to reconstruct substantial features like deep basin exchange flows, boundary currents, upwelling and downwelling, etc. Please provide solid evidences if you do need to keep this sentence.

Line 97. These indicated hotspots are reasonable and expectable Have the authors checked if these hotspots also exist in the other models like HYCOM? If they do exist, more discussions should be added that why those hotspots do not support CAC circulation in those models. If not, additional figures to clarify these concerns would be better to be provided.

Line 116 'Mindoro' -> 'Mindoro Strait'

Line 122-123 I do not see the 1500 and 3000 m isobaths indicated in the Figure.

Line 456-461 These references should be moved upward.

Response to reviewers

Reviewer #1 (Remarks to the Author):

The author has answered my questions and made necessary modifications to the manuscript. My suggestion is acceptable.

Response: Thanks for the reviewer's help for the improvement of the manuscript.

Reviewer #2 (Remarks to the Author):

In their revised version of the manuscript of 'Hotspots of the Stokes Rotating Circulation in a Large Marginal Sea', I am pleased that the authors responded properly to most of our reviews. This manuscript may be accepted after some minor corrections as below.

Response: Appreciate the reviewer's careful review and suggestion for the improvement of the manuscript.

Line 40-41. Except for mentioning some other marginal seas when addressing the background, no more case study has been done outside of the SCS by this study. So, I suggest either add case study of other marginal seas, or remove this sentence.

Response: The sentence is removed as suggested.

Fig.1a&b. The model based on which these results are calculated should been indicated. E.g., '(a) Schematic annual mean cyclonic-anticyclonic-cyclonic ... of the SCS, respectively, based on CMOMS'.

Response: Done.

Line 73 'than' -> 'with'

Response: Done.

Line 78 'through the Luzon Strait'

Response: Done.

Line 88-89. In my opinion, these models do not manage to simulate the intermediate and deep circulation well in the adjacent oceans. A lot discrepancies exist among these models and most of them fail to reconstruct substantial features like deep basin exchange flows, boundary currents, upwelling and downwelling, etc. Please provide solid evidences if you do need to keep this sentence.

Response: The global models perform relatively well in the adjacent open ocean, as compared with their performance in the marginal sea. To avoid confusion, the sentence is removed.

Line 97. These indicated hotspots are reasonable and expectable Have the authors checked if these hotspots also exist in the other models like HYCOM? If they do exist, more discussions

should be added that why those hotspots do not support CAC circulation in those models. If not, additional figures to clarify these concerns would be better to be provided.

Response: The flow pattern of CAC circulation and hotspots are twin, **and** since other models like HYCOM have not captured the layered CAC circulation in the SCS (Fig. 1c), we think that it may not be logically sensible to discuss hotspots (of these models) which compose the CAC circulation.

Regarding to the reasons for the absence of the CAC circulation in other models, it requires comprehensive investigation in a separated study. We demonstrated this important issue of global models by conducting “hotspots” analyses in this paper.

As pointed out in the last Response:

“There are many reasons for this deficiency in the global models, as pointed out in the paper. A better understanding and better representation of the complex circulation physics are certainly critical to improving of regional simulations in the global models, and it is the aim of this study to provide both.

Although it would require a separated study to thoroughly provide answer for the failure of global models in capturing the CAC circulation in the SCS, we summarize the following major reasons as presented in the paper:

1: The resolution is not a major issue since the global circulation models, such as CMEMS24, OFES25, HYCOM26, and others (not shown), have the same or higher resolutions as compared with our model, as mentioned in the paper.

2: Better representation of the complex marginal sea physics is critical, such as flow-topography interaction internally and strait dynamics externally as demonstrated by hotspots dynamics in this study. The ocean circulation in marginal sea is forced not only by wind and intrusion from the adjacent seas externally, but also by intensive dynamics processes involving flow-topography interaction internally. More importantly, the internal and external forcings couple each other to control both the circulation in marginal sea and water exchange between the sea and adjacent ocean. The global models did not well simulate the exchanges between the SCS and the adjacent oceans.

The global models did not well capture such as slope topography effect of the marginal sea and straits linking the sea and adjacent ocean. These may be attributed to, for example, misrepresentation of topography by vertical coordinate (e.g. by z-coordinate), twisted circulation dynamics by over-usage of data assimilation, and incoherent dynamics linkage among the adjacent ocean and sea. A physics and physical-accommodated investigation of global models require a separate investigation.”

Line 116 ‘Mindoro’ -> ‘Mindoro Strait’

Response: Done.

Line 122-123 I do not see the 1500 and 3000 m isobaths indicated in the Figure.

Response: We change the caption of figure to make it clear. ‘ The isobaths of the 200 m, 500 m, 1000 m, 1500 m and 2000 m in the upper layer in (a) indicate the location of the slope. The 1500 m and 3000 m isobaths are shown in the middle and deep layers in (b), respectively. ‘

The arrangement avoids too many contours and too busy in the figure,

Line 456-461 These references should be moved upward.

Response: We follow the format of the journal.